# CEGA: A Cost-Effective Approach for Graph-Based Model Extraction and Acquisition

Zebin Wang [1]  Menghan Lin [2]  Bolin Shen [3]  Ken Anderson [3]  Molei Liu [4]  Tianxi Cai [1]  Yushun Dong [3]

## Abstract

Graph Neural Networks (GNNs) have demonstrated remarkable utility across diverse applications, and their growing complexity has made Machine Learning as a Service (MLaaS) a viable platform for scalable deployment. However, this accessibility also exposes GNN to serious security threats, most notably model extraction attacks (MEAs), in which adversaries strategically query a deployed model to construct a high-fidelity replica. In this work, we evaluate the vulnerability of GNNs to MEAs and explore their potential for cost-effective model acquisition in non-adversarial research settings. Importantly, adaptive node querying strategies can also serve a critical role in research, particularly when labeling data is expensive or time-consuming. By selectively sampling informative nodes, researchers can train high-performing GNNs with minimal supervision, which is particularly valuable in domains such as biomedicine, where annotations often require expert input. To address this, we propose a node querying strategy tailored to a highly practical yet underexplored scenario, where bulk queries are prohibited, and only a limited set of initial nodes is available. Our approach iteratively refines the node selection mechanism over multiple learning cycles, leveraging historical feedback to improve extraction efficiency. Extensive experiments on benchmark graph datasets demonstrate our superiority over comparable baselines on accuracy,

fidelity, and F1 score under strict query-size constraints. These results highlight both the susceptibility of deployed GNNs to extraction attacks and the promise of ethical, efficient GNN acquisition methods to support low-resource research environments. Our implementation is publicly available at https://github.com/LabRAI/CEGA.

## 1. Introduction

Graph Neural Networks (GNNs) have achieved remarkable performance in a variety of applications powered by graph learning, such as molecular graph structure analysis (Sun et al., 2022; Wang et al., 2023; Zang et al., 2023; Zhao et al., 2024), fraud detection (Qin et al., 2022; Cheng et al., 2024; Motie & Raahemi, 2024; Lou et al., 2025), and healthcare diagnostics (Ahmedt-Aristizabal et al., 2021; Lu & Uddin, 2021; Zafeiropoulos et al., 2023; Paul et al., 2024). However, as GNNs grow in complexity and computational demands, training them from scratch becomes increasingly prohibitive due to rising computational costs (Abbahaddou et al., 2024; Kose et al., 2024). To address this, graph-based Machine Learning as a Service (MLaaS) has emerged as a cost-effective alternative, allowing users to access powerful pre-trained GNN models via APIs provided by service providers (Liu et al., 2022; Wu et al., 2023a; 2024).

Nevertheless, despite the advantages of graph-based MLaaS, such an inference paradigm also exposes GNN models to serious security risks, with GNN-based model extraction attacks (MEAs) posing a particularly significant threat (Wu et al., 2023a; 2024). Specifically, the goal of a model extraction attacker is to replicate the functionality of a GNN model owned by the service provider (i.e., the target model) by strategically querying it and using the responses to construct a local replica (i.e., the extracted model) (Shen et al., 2022; Wu et al., 2022). Such graph-based MEAs can lead to severe consequences such as copyright violations and patent infringement, especially in high-stake applications. For example, in the pharmaceutical industry, GNNs are widely used to predict molecular-level drug-target interactions (DTIs) (Wieder et al., 2020; Zhang et al., 2022; Tran et al., 2022). In this context, graph-based MLaaS provides

[1] Department of Biostatistics, T. H. Chan School of Public Health, Harvard University, Boston, Massachusetts, USA [2] Department of Statistics, Florida State University, Tallahassee, Florida, USA [3] Department of Computer Science, Florida State University, Tallahassee, Florida, USA [4] Department of Biostatistics, Mailman School of Public Health, Columbia University, New York, New York, USA. Correspondence to: Tianxi Cai <tcai@hsph.harvard.edu>, Yushun Dong <yd24f@fsu.edu>.

*Proceedings of the 42nd International Conference on Machine Learning*, Vancouver, Canada. PMLR 267, 2025. Copyright 2025 by the author(s).

pharmaceutical companies with a cost-effective and efficient platform for conducting related studies (Ahmedt-Aristizabal et al., 2021; Lu & Uddin, 2021; Vora et al., 2023). However, MEAs targeting such GNNs pose a serious risk to proprietary data, threatening trade secrets and potentially enabling unauthorized redistribution and unfair competition. These concerns may ultimately result in substantial financial and reputational damage (Bessen & Meurer, 2008; Nealey et al., 2015; Armstrong, 2016). Consequently, appropriately understanding and managing the threat of MEAs against GNNs has become a pressing concern (Zhao et al., 2025).

Beyond malicious uses, the research-driven acquisition of GNN functionality offers significant value, particularly for tailoring models to specialized downstream applications. A compelling example is the analysis of knowledge graphs (KGs) constructed from electronic health records (EHRs), where nodes represent clinical concepts—such as diagnoses, medications, and procedures—and edges capture relationships based on medical ontologies or empirical patterns of co-occurrence (Wang et al., 2014; Hong et al., 2021). In EHR-based KG research, deploying graph-based models for specific inference tasks within local health systems is often hampered by practical constraints, including the incompleteness of local database, limitations on data sharing across institutions, and heterogeneity in clinical practice patterns and patient populations, which can significantly limit model generalizability (Zhou et al., 2022; 2025). In such settings, effectively extracting and acquiring a well-trained target model developed on large-scale EHR data presents a powerful alternative to the costly and often impractical process of training from scratch using local EHR data (Lin et al., 2023a; Gan et al., 2025). This strategy not only improves computational efficiency but also enables advanced applications such as non-linear statistical inference on clinical knowledge graphs and ontology-informed learning for downstream applications (Xu et al., 2023; Liu et al., 2024).

In response to the pressing need outlined above, it is essential to systematically investigate strategies for extracting and acquiring graph-based model functionality. On the one hand, such efforts enable rigorous assessment of the severity of MEA threats to MLaaS platforms and inform the development of robust defense mechanisms (Zhang & Zitnik, 2020; Mujkanovic et al., 2022; Ennadir et al., 2023; Dong et al., 2024; Cheng et al., 2025). On the other hand, they also support the efficiency and feasibility of research-oriented GNN acquisition, as demonstrated in recent work on surrogate learning and transfer-based GNN extraction (Huo et al., 2023; Oloulade et al., 2023). However, despite the urgency and potential impact of such research, designing systematic strategies for extracting and acquiring GNN functionality remains a non-trivial task. In particular, we face two fundamental challenges:

(1) *Stringent budget and query batch size constraints.* First, excessive querying incurs substantial computational and financial costs under the pay-per-query basis, making large-scale extraction on well-trained MLaaS models economically unfeasible (Hou et al., 2019; Gong et al., 2020; Wu et al., 2023b). Second, querying in bulky batches risks violating MLaaS user agreements or triggering security alerts, as many providers implement monitoring mechanisms to identify and block potentially adversarial queries (Brundage et al., 2018; Juuti et al., 2019).

(2) *Structural dependency between nodes.* First, nodes can naturally exhibit various types of dependencies between each other in real-world graphs, depending on what types of semantics the edges encode (Zhou et al., 2020; Wu et al., 2021). Second, these dependencies across a broad localized area in the graph topology can collectively influence the information that the extracted model can acquire (Ju et al., 2024; Kahn et al., 2025).

Multiple research works have taken early steps to explore model extraction against GNNs for node-level graph learning tasks (DeFazio & Ramesh, 2019; Shen et al., 2022; Wu et al., 2022). However, these studies overlook the practical constraints on budget and batch size. More recently, several research works attempted to handle query budget limitations on MEAs (Shi et al., 2017; Liu et al., 2023; Karmakar & Basu, 2023). However, these approaches can hardly be generalized to graph learning tasks, as they often overlook the fact that GNNs can embed deeper information to the graph data during processing, even when certain features are absent or filtered out from the input (Dong et al., 2025). Therefore, the study of addressing the two practical challenges above specifically tailored for node-level graph learning tasks remains nascent.

To address the aforementioned challenges, we propose a targeted approach for the extraction and acquisition of GNN functionality, termed Cost-Efficient Graph Acquisition (CEGA). Our framework is specifically designed to balance effectiveness and efficiency in acquiring GNNs under realistic constraints. Without loss of generality, we focus on GNNs performing node classification, one of the most widely studied and fundamental tasks in node-level graph learning. Specifically, to overcome the challenge of budget and query batch size constraints, CEGA is designed to incorporate historical information from the initial and previous queries, starting with a very limited number of queries, in each of its iterations to improve its informativeness in node selection. To overcome the challenge of structural dependency between nodes, we prioritize nodes with high structural centrality to ensure queries can capture information that aligns with the localized graph topology. Furthermore, we introduce a diversity metric to prevent query clustering at the structural level and improve stability. Extensive

empirical experiments on real-world benchmark datasets demonstrated the superiority of the proposed CEGA framework in extracting the target GNN models under realistic query constraints, delivering key practical significance over existing alternatives.

In summary, the contributions of this paper are three-fold:

- **Novel Problem Formulation**: We introduce the problem of *GNN Model Extraction With Limited Budgets* specifically within the context of node-level graph learning tasks. This formulation offers a more realistic and practical setting for GNN model extraction compared to prior work in model extraction.

- **Comprehensive Methodology Design**: We present a novel framework for GNN model extraction and acquisition that dynamically identifies the most informative queries throughout the training process. Our approach integrates guidance based on three complementary criteria: *representativeness*, *uncertainty*, and *diversity*, enabling efficient and effective query selection.

- **Extensive Empirical Evaluation**. We conduct comprehensive experiments on real-world graph datasets to demonstrate the effectiveness of the proposed CEGA framework. Evaluation metrics include both model faithfulness and downstream utility, showing CEGA's superiority over existing alternatives.

## 2. Preliminaries

**Notations.** Suppose that we have a GNN model $f_\mathrm{T}$ trained on the target graph $\mathcal{G}_\mathrm{T} = \{\mathcal{V}_\mathrm{T}, \mathcal{E}_\mathrm{T}\}$, where $\mathcal{E}_\mathrm{T}$ denotes the edges of $\mathcal{G}_\mathrm{T}$. In the acquisition process, we assume knowledge to a pool of candidate nodes for querying, denoted as $\mathcal{V}_\mathrm{a}$, and a respective graph structure, denoted as $\mathcal{G}_\mathrm{a}$. We consider an iterative node querying approach with $\Gamma$ learning cycles in total. We denote the initial query set as $\mathcal{V}_0$, with budget $\mathcal{I} = |\mathcal{V}_0|$. On the $\gamma$th iterative cycle with $\gamma \in \{1, 2, ..., \Gamma\}$, we use $\mathcal{V}_{\gamma-1}$ to denote the collection of nodes queried in previous cycles, where $\mathcal{V}_0 \subsetneq \mathcal{V}_1 \subsetneq \mathcal{V}_2 \subsetneq ... \subsetneq \mathcal{V}_\Gamma \subsetneq \mathcal{V}_\mathrm{a}$. In this cycle, we query $\kappa$ nodes from the candidate set $\mathcal{V}_\mathrm{a} \backslash \mathcal{V}_{\gamma-1}$, with capacity $n_{\gamma-1} = |\mathcal{V}_\mathrm{a} \backslash \mathcal{V}_{\gamma-1}|$. The attributes of the nodes belonging to $\mathcal{V}_\mathrm{a} \backslash \mathcal{V}_{\gamma-1}$ are denoted as $\mathcal{X}_{\gamma-1} = \{\mathbf{x}_{\gamma-1}^{(1)}, \mathbf{x}_{\gamma-1}^{(2)}, ..., \mathbf{x}_{\gamma-1}^{(n_{\gamma-1})}\}$, where each $\mathbf{x}_{\gamma-1}^{(j)}$ is an attribute vector with $d$-dimensions. For convenience, we denote the respective attribute of some node $v \in \mathcal{V}_\mathrm{a} \backslash \mathcal{V}_{\gamma-1}$ as $\mathbf{x}_{\gamma-1}^{(v)}$. The respective outcome for node $v$ in a model $f$ is denoted as $\widehat{\mathbf{y}}_v = f(\mathbf{x}^{(v)}, \mathcal{G}_\mathrm{a}) = \{\widehat{y}_v^{(1)}, \widehat{y}_v^{(2)}, ..., \widehat{y}_v^{(C)}\}$, where $\widehat{\mathbf{y}}_v$ is a probability vector of length $C$ representing the softmax scores for each class. Our notation system is compatible with existing works in the context of MEAs against GNNs, as highlighted by existing work such as (Wu et al., 2022; Shen et al., 2022; Wu et al., 2023a).

**Background.** Existing research works have rarely discussed GNN-based MEA contexts in real-world settings. In our paper, we expect to gain practical significance by considering a realistic setup to extract GNNs. In our setting, we pose upper limits on (1) the initial query budget $\mathcal{I}$, (2) the per-cycle query budget $\kappa$, and (3) the overall budget $B$. Our setting is more realistic compared with existing ones that rely on simultaneous high-volume queries since excessive queries impose significant costs, and frequent large-scale queries are likely to alert the maintainers of the target model. We focus on node classification tasks, which are among the most widely studied problems in node-level graph learning, as previously investigated by (Dong et al., 2023; Luan et al., 2023; Zhang et al., 2023; Li et al., 2024).

**Goal of Acquisition.** The researchers aim to extract a model $f_\mathrm{a}$ that closely replicates the behavior of target GNN model $f_\mathrm{T}$ using a limited number of queries constrained by an initial query budget $\mathcal{I}$, a per-cycle query budget $\kappa$, and an overall query budget $B$. Here, the similarity in their behavior is generally measured by the ratio of the same input-output pairs.

We then formulate the problem of *GNN Model Extraction With Limited Budgets* in Problem 1.

**Problem 1.** *(GNN Model Extraction With Limited Budgets). Given a target GNN model $f_\mathrm{T}$ and available prior knowledge $\{\mathcal{X}_\mathrm{a}, \mathcal{G}_\mathrm{a}\}$, the objective is to achieve an extracted model $f_\mathrm{a}$ with behaviors being as similar to $f_\mathrm{T}$ as possible for any given test node $v \in \mathcal{V}_\mathrm{T}$, while adhering to the constraints of limited initial query budget $\mathcal{I}$, per-cycle query budget $\kappa$, and total query budget $B$.*

## 3. Methodology

### 3.1. Overview

In this section, we introduce CEGA, an active sampling framework designed to extract and acquire GNN behaviors efficiently. CEGA employs a multilevel analysis strategy that iteratively selects informative nodes by leveraging prior heuristics derived from the initial query set $\mathcal{V}_0$ and the $\gamma - 1$ batches of previously queried nodes. these historical insights are summarized by an interim model $f_{\gamma-1}$, which guides the selection process in iteration $\gamma$.

Specifically, CEGA is designed to conduct node selection by incorporating three key objectives: (1) *Representativeness:* The queried nodes should capture the structural essence of the graph, facilitating an accurate reconstruction of model behavior across the network. (2) *Uncertainty:* Nodes with high uncertainty, as indicated by historical predictions, are prioritized, as they likely reside near decision boundaries of the interim prediction model $f_\gamma$. (3) *Diversity:* To avoid excessive clustering, selected nodes should be diverse in their distribution across the graph, ensuring a comprehensive

exploration of the underlying structure.

To achieve these goals, CEGA is equipped with three objectives, $\mathcal{L}_1^\gamma(v, \mathcal{G}_a)$, $\mathcal{L}_2^\gamma(v, \mathcal{G}_a)$, and $\mathcal{L}_3^\gamma(v, \mathcal{G}_a)$, evaluating the tendency of selecting a node $v$ from the candidate set $\mathcal{V}_a \backslash \mathcal{V}_{\gamma-1}$ in each querying cycle $\gamma$. Nodes are adaptively ranked and selected based on their combined ranking across these three criteria, ensuring an efficient and cost-effective querying strategy.

### 3.2. The Proposed Framework of CEGA

The CEGA framework begins by building a primitive initial GNN-based prediction model $f_0$ with $\mathcal{I}$ initial queried nodes. In each learning cycle, CEGA selects $\kappa$ nodes with the highest comprehensive rank through an adaptive node selection method based on the representativeness, uncertainty, and diversity of the nodes. A new interim model $f_\gamma$ involving all nodes queried in the past and new nodes selected for query in the same cycle is trained as a summarization of existing historical information to guide further queries. We perform such cycles iteratively until the budget limit $B$ is reached. Finally, we evaluate the performance of CEGA by training GNN models with queried nodes.

**Initialization.** In CEGA, we randomly select $\mathcal{I}$ initial nodes from the node pool for acquisition $\mathcal{V}_a$. A random selection of initial nodes reduces systematic bias and ensures comparability between CEGA and other approaches mentioned in the existing literature (Cai et al., 2017; Zhang et al., 2021).

**Graph Structure-Based Analysis for Representativeness.** To ensure that the queried nodes in each cycle are representative of the overall graph structure, we rank nodes based on structural indices that capture their relative importance within the network. Among such indices, we specifically incorporate PageRank (Page et al., 1999), where the objective function $\mathcal{L}_1^\gamma$ is given as

$$\mathcal{L}_1^\gamma(v, \mathcal{G}_a) = \frac{1-\xi}{N} + \xi \sum_{w \in \text{in}(v)} \frac{\mathcal{L}_1^\gamma(w, \mathcal{G}_a)}{L(w)}. \qquad (1)$$

In (1), $N = |\mathcal{V}_a|$ represents the total number of nodes in the extraction subgraph $\mathcal{G}_a$, $\text{in}(v)$ represents the collection of nodes with edges pointing to node $v$, $L(w)$ represents the number of outbound edges from node $w$, and $\xi$ represents a damping factor typically set at 0.85. The evaluation of $\mathcal{L}_1^\gamma$ is inherently recursive in computation, relying on iterative processes to update the scores of the nodes given their neighborhood until convergence. The rank of nodes according to their representativeness in the $\gamma$th cycle is denoted as $\mathcal{R}_1^\gamma$.

**History-Based Analysis for Uncertainty.** To evaluate the uncertainty of nodes in an interim GNN model $f_{\gamma-1}$ on the classification task, we evaluate the entropy of the nodes in the pool $\mathcal{V}_a \backslash \mathcal{V}_{\gamma-1}$ as their sensitivity against changes in the attributes of its neighbors. In particular, we give the

objective function as

$$\mathcal{L}_2^\gamma(v, \mathcal{G}_a) = -\sum_{i=1}^{C} \widehat{y}_{v;\gamma-1}^{(i)} \log(\widehat{y}_{v;\gamma-1}^{(i)}). \qquad (2)$$

In (2), $\widehat{y}_{v;\gamma-1}^{(i)}$ is the $i$th entry of $\widehat{\mathbf{y}}_{v;\gamma-1} = f_{\gamma-1}(\mathbf{x}_{\gamma-1}^{(v)}, \mathcal{G}_a)$. For downstream tasks that are less sensitive to time and space complexity, we propose a theory-backed alternative ranking mechanism that measures a node's resilience in maintaining its predicted label under moderate Gaussian perturbation. In practice, we consider a series of perturbation $\boldsymbol{\tau}^{(j)} \overset{i.i.d}{\sim} \mathcal{N}(\mathbf{0}, \epsilon^2 \mathbf{I})$ where $\mathbf{I} \in \mathbb{R}^{d \times d}$ is an identity matrix, and obtain the perturbed version of the attributes $\mathcal{X}_{\gamma-1}$, denoted as

$$\mathcal{T}_{\gamma-1} = \{\mathbf{x}_{\gamma-1}^{(1)} + \boldsymbol{\tau}^{(1)}, \mathbf{x}_{\gamma-1}^{(2)} + \boldsymbol{\tau}^{(2)}, ..., \mathbf{x}_{\gamma-1}^{(n_{\gamma-1})} + \boldsymbol{\tau}^{(n_{\gamma-1})}\}.$$

We repeat the perturbation $S$ times and obtain the perturb data $\mathcal{T}_{\gamma-1}^\ell$ where $\ell \in [S]$. For any node $v$, the respective probability for $\mathcal{T}_{\gamma-1}^\ell$ is denoted as $\widehat{\mathbf{y}}_{v;\gamma-1}^\ell = f_{\gamma-1}(\mathbf{x}_{\gamma-1}^{(v)} + \boldsymbol{\tau}_\ell^{(v)}, \mathcal{G}_a)$, where $\ell \in [S]$. The alternative objective function $\mathcal{L}_{2;\text{alt}}^\gamma(v, \mathcal{G}_a)$ is given as

$$\mathcal{L}_{2;\text{alt}}^\gamma(v, \mathcal{G}_a) = \sum_{\ell=1}^{S} \mathbb{I}_{\left\{ \arg\max\{\widehat{\mathbf{y}}_{v;\gamma-1}\} = \arg\max\{\widehat{\mathbf{y}}_{v;\gamma-1}^\ell\} \right\}}, \qquad (3)$$

where $\widehat{\mathbf{y}}_{v;\gamma-1}$ and $\widehat{\mathbf{y}}_{v;\gamma-1}^\ell$ are outputs of $f_{\gamma-1}$, which takes the subgraph $\mathcal{G}_a$ as an input. The rank of nodes based on their uncertainty on the prediction of the interim model $f_{\gamma-1}$ in the $\gamma$th cycle of CEGA is denoted as $\mathcal{R}_2^\gamma$.

In Section 3.3, we provide theoretical insights into the time and space complexity of CEGA to further justify its suitability to measure uncertainty in node selection. Furthermore, we present the theoretical guarantee for the existence of an appropriate perturbation parameter $\epsilon$ involved in the alternative approach. The parameter $\epsilon$ is expected to be sufficiently large to capture the sensitivity of the interim model's predictions while remaining small enough to preserve the stability and effectiveness of the interim model.

**Distance-Based Analysis for Diversity.** Finally, we evaluate the diversity of a node in the pool $\mathcal{V}_a \backslash \mathcal{V}_{\gamma-1}$ compared to the queried nodes $\mathcal{V}_{\gamma-1}$. To start, we apply the $K$-Means algorithm for the embedding of queried nodes with $K = C$, where $C$ is the number of classes for the graph dataset. We then compare the embeddings of the nodes belong to $\mathcal{V}_a \backslash \mathcal{V}_{\gamma-1}$ with the clusters formed by the queried nodes to determine whether they align with a category that is over-represented in $\mathcal{V}_{\gamma-1}$. To do this, we measure the distance between some node $v \in \mathcal{V}_a \backslash \mathcal{V}_{\gamma-1}$ and the $C$ centroids. We assign node $v$ to the class $j \in \{1, 2, ..., C\}$ such that the distance between its embedding $\mathcal{E}_v$ and the centroid $\mathcal{C}_j$ is

minimized. Here, $\mathcal{E}_v$ is an output of the interim model $f_{\gamma-1}$, with the graph structure $\mathcal{G}_a$ serving as a necessary input. We then establish the objective function $\mathcal{L}_3^\gamma(v, \mathcal{G}_a)$ as

$$\mathcal{L}_3^\gamma(v, \mathcal{G}_a) = \rho\varphi_{[0,1]}\left(\frac{1}{1+\delta_v}\right) + (1-\rho)\varphi_{[0,1]}\left(\frac{1}{1+|\mathcal{Q}_v|}\right). \tag{4}$$

Here $\delta_v$ is the minimal distance between the embedding $\mathcal{E}_v$ for node $v$ and centroids $\mathcal{C}_1, \mathcal{C}_2, ..., \mathcal{C}_C$, and we have

$$\delta_v = \min_{c\in\{1,2,...,C\}} \left\|\mathcal{E}_v - \mathcal{C}_c\right\|_2.$$

On the other hand, $\mathcal{Q}_v$ represents the collection of queried nodes that belong to the same centroid $\mathcal{C}_{c^*}$ as node $v$, where

$$c^* = \arg\min_{c\in\{1,2,...,C\}} \left\|\mathcal{E}_v - \mathcal{C}_c\right\|_2.$$

The number of nodes belong to $\mathcal{Q}_v$ is denoted as $|\mathcal{Q}_v|$. $\varphi_{[0,1]}(\cdot)$ represents the min-max scaling function. $\rho$ is a hyperparameter subject to tuning. The rationale behind the setup of $\mathcal{L}_3^\gamma$ is to guide the selection of nodes from $\mathcal{V}_a\backslash\mathcal{V}_{\gamma-1}$ that represent the embedding patterns of labels that are underrepresented in the queried nodes. By using $\mathcal{L}_3^\gamma$, we rank the nodes $v \in \mathcal{V}_a\backslash\mathcal{V}_{\gamma-1}$ in the order $\mathcal{R}_3^\gamma$.

**Adaptive Node Selection Method.** Once we obtain the ranking of all candidate nodes according to the objective functions $\mathcal{L}_1^\gamma$, $\mathcal{L}_2^\gamma$, and $\mathcal{L}_3^\gamma$, we compute a weighted average ranking of the nodes in the three categories and query the top-$\kappa$ nodes $\mathcal{V}_\gamma^Q$ based on this weighted average, as those nodes are expected to guide further informative queries to the target GNN model.

The weighted average ranking for cycle $\gamma$ is expressed as

$$\mathcal{R}^\gamma = \omega_1(\gamma)\mathcal{R}_1^\gamma + \omega_2(\gamma)\mathcal{R}_2^\gamma + \omega_3(\gamma)\mathcal{R}_3^\gamma.$$

The cycle-specific weights $\omega_1$, $\omega_2$, and $\omega_3$ are subject to adaptive optimization according to the principle, as inspired by (Cai et al., 2017), that the representativeness rank $\mathcal{R}_1$ does not rely on the interim model $f_{\gamma-1}$, while $\mathcal{R}_2$ and $\mathcal{R}_3$ rely on the $f_{\gamma-1}$. The weight $\omega_1$ is assigned a higher value when $\gamma$ is small, reflecting the relatively poor performance of the interim model during the earlier stages of querying. As $\gamma$ increases, $\omega_2$ and $\omega_3$ are progressively raised, resonating the improved performance of the interim model in later querying cycles. The dynamic node selection ensures that the weights can fit CEGA's progressive querying process.

**Learning to Guide Queries and Output.** After obtaining queried nodes $\mathcal{V}_\gamma$ in the $\gamma$th cycle, we train the new interim model $f_\gamma$ based on $\{\mathcal{V}_\gamma, \mathcal{G}_a\}$ and the previous interim model $f_{\gamma-1}$ for $E$ epochs. The updated model $f_\gamma$ then guides node selection for further queries in the $(\gamma + 1)$th cycle. After completion of the $\Gamma$ querying cycles, CEGA returns the collection of queried nodes $\{\mathcal{V}_1, \mathcal{V}_2, ..., \mathcal{V}_\Gamma\}$. We summarize the algorithmic routine of CEGA in Algorithm 1.

---

**Algorithm 1** The Proposed Framework of CEGA

Initialization: Query initial nodes $\mathcal{V}_0$, where $|\mathcal{V}_0| = \mathcal{I}$, from $\mathcal{V}_a$.
Train the initial model $f_0$ on $\{\mathcal{V}_0, \mathcal{G}_a\}$.
**for** Cycle $\gamma$ from 1 to $\Gamma$ **do**
  **if** $\mathcal{I} + (\gamma - 1)\kappa < B$ **then**
    Evaluate the representativeness score $\mathcal{L}_1^\gamma$, uncertainty score $\mathcal{L}_2^\gamma$, and diversity score $\mathcal{L}_3^\gamma$ for all candidate nodes in $\mathcal{V}_a\backslash\mathcal{V}_{\gamma-1}$.
    Obtain node ranks $\mathcal{R}_1^\gamma$, $\mathcal{R}_2^\gamma$, and $\mathcal{R}_3^\gamma$.
    Select and query top-$\kappa$ nodes $\mathcal{V}_\gamma^Q$ via the adaptive selection method.
    Set $\mathcal{V}_\gamma = \mathcal{V}_{\gamma-1} \cup \mathcal{V}_\gamma^Q$.
  **else**
    Set $\mathcal{V}_\gamma = \mathcal{V}_{\gamma-1}$.
  **end if**
  Train the new interim model $f_\gamma$ based on $\{\mathcal{V}_\gamma, \mathcal{G}_a\}$ and $f_{\gamma-1}$ for $E$ epochs.
**end for**
Return the nodes collection $\{\mathcal{V}_1, \mathcal{V}_2, ..., \mathcal{V}_\Gamma\}$.

---

### 3.3. Theoretical Analysis

In this section, we summarize the theoretical results for CEGA's measurements with respect to the uncertainty of candidate nodes indicated by history-inspired interim models. As outlined in Section 3.2, we address two core aspects: CEGA's efficiency in referring to the history guide and the existence of an appropriate perturbation parameter $\epsilon$ for the alternative. The detailed proofs are given in Appendix A.

First, we provide a thorough analysis of the time and space complexity of CEGA's uncertainty measurement approach. Proposition 3.1 highlights the feasibility of our approach, indicating its resource-friendly feature required for graph data with complex attributes and subgraph structure.

**Proposition 3.1** (Evaluation of Complexity). *Suppose that the base model of CEGA is a $L$-layer GCN. Under the conditions such that*

1. *The number of nodes queried by CEGA in each cycle, indicated by $\kappa$, is $\Theta(1)$;*
2. *$d \gg h = \Theta(C)$, where $d$ indicates the dimension of the attributes for the graph data, $h$ indicates the dimension of the node embeddings, and $C$ indicates the number of classes in the softmax score output,*

*CEGA's entropy-based approach introduces an additional time complexity of $O(CN + N \log N)$ and space complexity of $O(CN)$, building on the $O(LN^2 d + LN d^2)$ time complexity and $O(N^2 + Ld^2 + LNd)$ space complexity required for CEGA to compute embeddings and softmax scores via forward propagation for the analysis in Section 3.2.*

**Remarks for Proposition 3.1.** CEGA's entropy-based approach provides superior scalability and adaptability for large, complex graph datasets, introducing minimal additional time and space complexity beyond attribute propagation through the interim model for embeddings and softmax scores. Notably, these added complexities are of significantly lower order than the training and forward propagation costs of the interim model.

In Theorem 3.2, we show the existence of an appropriate perturbation intensity $\epsilon$ that is sufficiently small to maintain the overall stability of the history guide $f_\gamma$. This ensures the feasibility of perturbation-based alternative uncertainty evaluation in CEGA by guaranteeing that the approach can reliably capture prediction uncertainty from history without causing inconsistency in the interim model, even when random noise is applied to the node attributes.

**Theorem 3.2** (Existence of Feasible Perturbation). *Consider the perturbation scheme for the evaluation of node uncertainty under the interim GNN model $f_\gamma$ in CEGA. We show that there exists some perturbation intensity $\epsilon$ such that $\left\| f_\gamma(\mathbf{x}_i, \mathcal{G}_a) - f_\gamma(\widetilde{\mathbf{x}}_i, \mathcal{G}_a) \right\|_2$ holds the stability conditions, where $\widetilde{\mathbf{x}}_\tau$ is the perturbation of $\mathbf{x}_\tau$ where $\widetilde{\mathbf{x}}_\tau - \mathbf{x}_\tau \sim \mathcal{N}(0, \epsilon^2)$. Specifically, for any $\zeta > 0$, there exists some $\epsilon = \epsilon(\zeta, \delta)$ such that $\left\| f_\gamma(\mathbf{x}_i, \mathcal{G}_a) - f_\gamma(\widetilde{\mathbf{x}}_i, \mathcal{G}_a) \right\|_2 \leq \zeta$ with probability at least $1 - O(\delta)$.*

## 4. Experimental Evaluation

In this section, we present an in-depth experimental evaluation of CEGA, addressing three key research questions: **RQ1**: How effective is CEGA in graph-based model extraction and acquisition tasks compared to baseline methods on a fixed querying budget? **RQ2**: How well does CEGA recover the target model on a limited query budget, as opposed to querying all nodes that can be queried? **RQ3**: What is the contribution of each evaluation module of CEGA to its overall performance?

### 4.1. Experimental Settings

**Graph Learning Task and Datasets.** We evaluate CEGA on the extraction task for graph-based node classification models, assuming that the extraction side has access to the attributes of the candidate nodes $\mathcal{V}_a$ and the structure of subgraph $\mathcal{G}_a$ that involves $\mathcal{V}_a$. This setup is categorized as *Attack 0* in MEA literature such as (Wu et al., 2022). Our experiments are conducted on 6 widely used benchmark datasets: (1) Coauthorship networks where nodes are authors and edges represent collaboration, including *Coauthor-CS* and *Coauthor-Physics*; (2) Co-purchase graphs with nodes as products and edges as items frequently purchased together, including *Amazon-Computer* and *Amazon-Photo*; and (3) Academic citation and collaboration network, including *Cora-Full* and *DBLP*. These datasets vary in size,

complexity, and formality of node attributes, providing a comprehensive basis for evaluating CEGA's performance. The dataset statistics are provided in Appendix B.1.

**Training Protocol.** In our experiment, we consider two models trained on the datasets of interest. The *full subgraph model* is trained on $\{\mathcal{V}_a, \mathcal{G}_a\}$, where the subgraph known to the extractors $\mathcal{G}_a$ satisfies $\mathcal{G}_a \subsetneq \mathcal{G}_T$, to establish the upper limit of performance that model extraction approaches can achieve in their respective task under our assumptions. In the *budget-constrained model* trained on $\{\mathcal{V}_\Gamma, \mathcal{G}_a\}$, where $|\mathcal{V}_\Gamma| \ll |\mathcal{V}_a|$ in many practices, the model extraction task is conducted on a more restrictive but realistic scenario with budget constraints, as we have defined in Section 2. In response to the constraints, CEGA and the baseline models, as detailed in the following sections, progressively select and query the most informative nodes from the pool of candidate nodes for optimal performance. To show the superiority of CEGA, we compare the performance of all models tested under a budget-constrained setup and examine the performance incrementation of the full subgraph model over the budget-constrained model for each approach.

**Evaluation Metrics.** We evaluate the accuracy and F1 score of budget-constrained models built under nodes selectively queried by all the tested approaches. Furthermore, we evaluate the faithfulness of these models to the target model $f_T$, using fidelity as the metric. The measurements on accuracy, F1 score, and fidelity are further compared between the full subgraph model and the budget-constrained model to highlight the relative efficiency of the strategies tested. Finally, we evaluate the mean and standard deviation of accuracy, fidelity, and F1 score for budget-constrained models trained on nodes queried using CEGA and its variants with certain components ablated. All empirical evaluations are based on consistent settings with commonly used ones. The query budget is set as the number of label classes for each graph dataset multiplied by a fixed factor, ranging from $2C$ to $20C$, following widely accepted prior work, such as (Yang et al., 2016; Cai et al., 2017; Zhang et al., 2021).

**Baselines.** In our experiment, the performance of CEGA is compared against the *Random* baseline, where all the queried nodes are selected randomly from $\mathcal{V}_a$. Furthermore, we compare CEGA with the state-of-the-art active learning (AL) techniques specially designed for GNN in a query-by-training process consistent with CEGA for fair comparison, including *AGE* ((Cai et al., 2017)), *GRAIN (NN-D)*, and *GRAIN (ball-D)* (Zhang et al., 2021). To ensure consistency, all baseline models adhere to the same query constraints initialization setup. Details on the hyperparameter setup for all baseline methods and the proposed framework CEGA are included in Appendix B.2.

*Table 1.* Test accuracy, fidelity, and F1 score on different datasets using $20C$ queried nodes. Dataset abbreviations: CoCS (Coauthor-CS), CoP (Coauthor-Physics), AmzC (Amazon-Computer), AmzP (Amazon-Photo), Cora-Full, and DBLP. All numerical values are reported in percentage. The best results are in **bold**.

|          |              | CoCS | CoP | AmzC | AmzP | Cora_Full | DBLP |
|----------|--------------|------|-----|------|------|-----------|------|
| **Accuracy** | **Random**       | $88.75 \pm 0.7$ | $91.50 \pm 1.3$ | $83.79 \pm 0.9$ | $90.15 \pm 2.6$ | $49.73 \pm 0.3$ | $69.14 \pm 1.9$ |
|          | **GRAIN(NN-D)**  | $89.77 \pm 0.6$ | $93.37 \pm 0.8$ | $83.89 \pm 1.7$ | $90.98 \pm 0.3$ | $51.57 \pm 1.0$ | $68.37 \pm 0.9$ |
|          | **GRAIN(ball-D)**| $89.43 \pm 0.6$ | $93.37 \pm 1.0$ | $82.48 \pm 2.1$ | $90.01 \pm 1.2$ | $51.27 \pm 1.3$ | $68.57 \pm 1.0$ |
|          | **AGE**          | $\mathbf{90.68 \pm 0.4}$ | $93.69 \pm 0.3$ | $85.13 \pm 0.6$ | $90.79 \pm 2.6$ | $50.59 \pm 0.3$ | $72.41 \pm 2.2$ |
|          | **CEGA**         | $90.57 \pm 0.5$ | $\mathbf{93.90 \pm 0.4}$ | $\mathbf{85.98 \pm 0.4}$ | $\mathbf{91.95 \pm 0.3}$ | $\mathbf{52.74 \pm 0.6}$ | $\mathbf{73.29 \pm 0.9}$ |
| **Fidelity** | **Random**       | $91.43 \pm 0.8$ | $93.15 \pm 1.4$ | $88.45 \pm 1.0$ | $93.31 \pm 2.7$ | $74.06 \pm 0.8$ | $73.86 \pm 2.2$ |
|          | **GRAIN(NN-D)**  | $92.41 \pm 0.8$ | $95.11 \pm 0.9$ | $88.65 \pm 2.0$ | $94.17 \pm 0.6$ | $76.65 \pm 1.6$ | $72.71 \pm 1.1$ |
|          | **GRAIN(ball-D)**| $92.00 \pm 0.7$ | $95.19 \pm 1.2$ | $86.89 \pm 2.4$ | $92.93 \pm 1.4$ | $76.18 \pm 1.5$ | $73.35 \pm 1.2$ |
|          | **AGE**          | $\mathbf{93.61 \pm 0.5}$ | $95.55 \pm 0.4$ | $90.10 \pm 0.7$ | $93.97 \pm 2.9$ | $75.67 \pm 0.8$ | $77.18 \pm 2.4$ |
|          | **CEGA**         | $93.40 \pm 0.6$ | $\mathbf{95.83 \pm 0.5}$ | $\mathbf{90.81 \pm 0.4}$ | $\mathbf{95.33 \pm 0.5}$ | $\mathbf{77.90 \pm 0.9}$ | $\mathbf{78.50 \pm 0.9}$ |
| **F1**   | **Random**       | $81.44 \pm 1.6$ | $87.70 \pm 2.4$ | $78.95 \pm 1.8$ | $86.54 \pm 5.3$ | $27.56 \pm 0.3$ | $57.46 \pm 5.0$ |
|          | **GRAIN(NN-D)**  | $85.61 \pm 1.4$ | $90.93 \pm 1.0$ | $80.29 \pm 3.5$ | $88.06 \pm 0.8$ | $28.93 \pm 1.0$ | $58.72 \pm 3.7$ |
|          | **GRAIN(ball-D)**| $85.38 \pm 0.9$ | $90.97 \pm 1.4$ | $74.47 \pm 5.9$ | $86.99 \pm 2.0$ | $28.62 \pm 1.1$ | $60.87 \pm 3.5$ |
|          | **AGE**          | $\mathbf{87.65 \pm 0.4}$ | $91.58 \pm 0.5$ | $78.37 \pm 3.5$ | $89.14 \pm 3.2$ | $29.28 \pm 0.5$ | $65.72 \pm 3.2$ |
|          | **CEGA**         | $87.41 \pm 0.5$ | $\mathbf{91.78 \pm 0.7}$ | $\mathbf{82.57 \pm 1.4}$ | $\mathbf{90.06 \pm 0.6}$ | $\mathbf{31.20 \pm 0.8}$ | $\mathbf{67.35 \pm 1.5}$ |

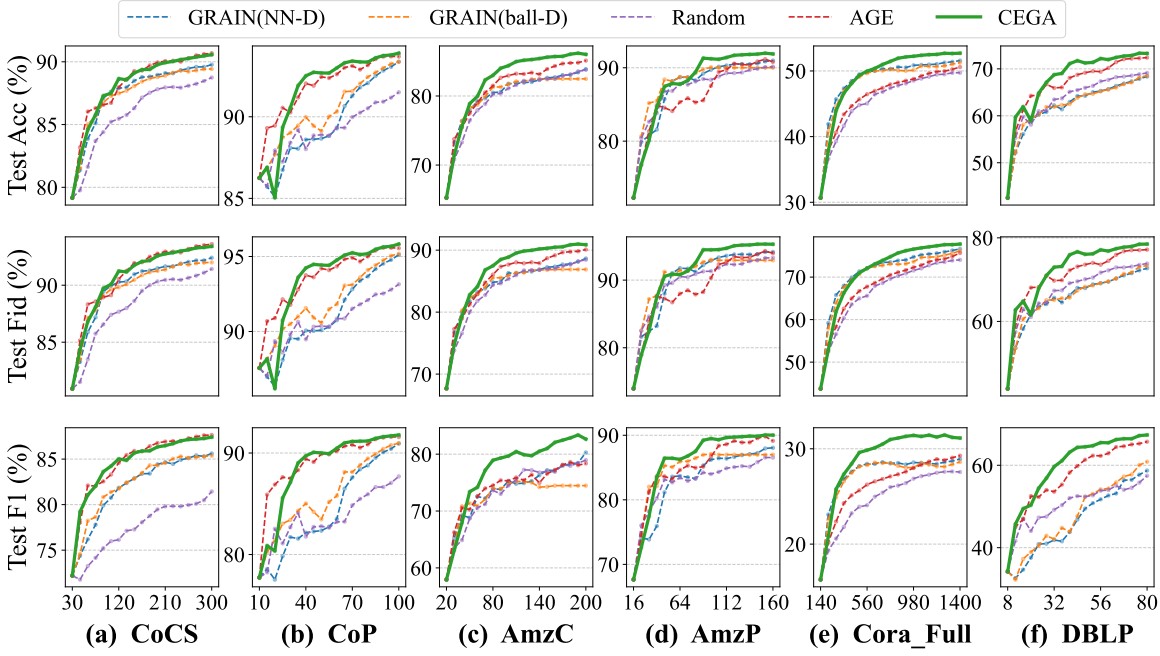

*Figure 1.* The trajectory of test accuracy, fidelity, and F1 score on different datasets using $2C$ to $20C$ queried nodes. The performance trajectory of CEGA is bolded in green, showing significant superiority over the alternatives across different number of queried nodes.

## 4.2. Evaluation on Budget-Constrained Model

To answer **RQ1**, we evaluate the performance of CEGA and various AL-based baseline methods on history-based progressive node querying in all the 6 datasets under different labeling budget scenarios. We incrementally raise the query budget from $2C$ to $20C$. Our evaluation primarily focuses on the fidelity of the budget-constrained subgraph model to the target model, while also considering accuracy and F1 score as supplementary metrics. To account for variability due to randomized initialization, each method is evaluated five times, and the mean performance is reported.

Figure 1 presents the trajectory of measured metrics with

the extracted model trained with $2C$ to $20C$ queried nodes. Table 1 presents a direct comparison between the performance of CEGA and the baseline models using $20C$ queried nodes in the 6 datasets of interest. We summarize the key observations below: (1) From the perspective of comparative performance, CEGA consistently achieves significant improvements in accuracy, fidelity, and F1 score across varying budget levels, demonstrating its capability to closely mimic the target model under stringent query budget constraints. CEGA also shows strong adaptability to the graph datasets being tested, which is considered one of its main advantages over the baselines. (2) From the perspective of the progressive nature of the models, as the budget increases from $2C$ to $10$-$15C$, CEGA further extends its advantage over baseline methods among all the metrics tested, particularly for the Amazon-Computer and Cora-Full datasets. This indicates that CEGA's node selection strategy effectively identifies the most informative nodes throughout the iterative querying process, leading to superior alignment with the target model as more queries become available.

### 4.3. Comparison between Full Subgraph Model and Budget-Constrained Model

To answer **RQ2**, we highlight the effectiveness of CEGA in detecting the most informative nodes by comparing the accuracy, fidelity, and F1 score recovery performance between the budget-constrained model and the full subgraph model, as specified in Section 4.1. Specifically, we define a *performance gap* as the performance discrepancy between the full subgraph model and budget-constrained models following different querying approaches.

We visualize the performance gap measured by accuracy, fidelity, and F1 score across all 6 datasets of interest in Figure 2. Further results on the performance gap are presented in Table 4 and discussed in Appendix B.3. We summarize the key observations below: (1) From the perspective of measurement, CEGA consistently exhibits a lower performance gap across all the metrics tested (accuracy, fidelity, F1 score) compared to the baselines, indicating its superior ability to recover as much information as possible under stringent budget constraints. (2) From the perspective of adaptability, CEGA maintains a consistently lower gap across all the datasets we tested by a notifiable margin (e.g., 1-2%) despite variations in node attributes and graph structures. This reveals that CEGA is more robust and effective than baselines in conducting model extraction and acquisition on graph data with different levels of intrinsic complexity.

### 4.4. Ablation Study

To answer **RQ3**, we perform an ablation study by systematically removing the contribution of each one out of the 3 evaluation modules of CEGA. We then compare the perfor-

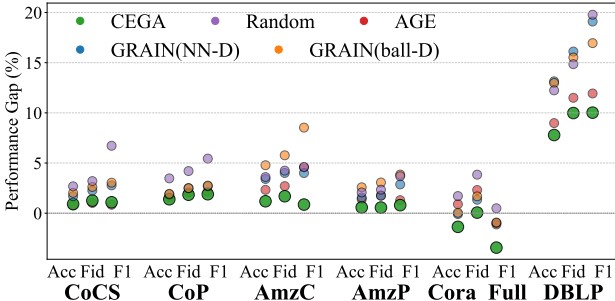

*Figure 2.* Model performance gaps between budget-constrained and full subgraph models, measured by accuracy, fidelity, and F1 score, across datasets. The gap indicates the negative impact of the budget constraints on the model performances. Therefore, lower gaps (i.e., less negative impact) are preferred.

*Table 2.* Ablation study results on fidelity for CEGA and variants with one evaluation module removed. *Cen* stands for Centrality, *UnC* stands for Uncertainty, *Div* stands for Diversity. The best results are in **bold**.

|            | CEGA            | No Cen         | No UnC         | No Div          |
|------------|-----------------|----------------|----------------|-----------------|
| **CoCS**   | **93.4 ± 0.6**  | 93.2 ± 0.2     | 91.9 ± 0.5     | 93.4 ± 0.6      |
| **CoP**    | **95.8 ± 0.5**  | 94.9 ± 0.4     | 90.2 ± 3.3     | 95.7 ± 0.5      |
| **AmzC**   | **90.8 ± 0.4**  | 90.0 ± 1.2     | 87.1 ± 2.2     | 90.7 ± 0.7      |
| **AmzP**   | **95.3 ± 0.5**  | 95.1 ± 0.3     | 93.7 ± 0.9     | 95.3 ± 0.7      |
| **Cora_Full** | 77.9 ± 0.9   | 75.3 ± 0.6     | 74.9 ± 0.9     | **78.3 ± 1.1**  |
| **DBLP**   | 78.5 ± 0.9      | 74.2 ± 2.4     | 65.1 ± 5.5     | **78.6 ± 1.4**  |

mance of the resulting models with only the two remaining modules involved with that of the full CEGA model. We evaluate budget-constrained models with a query budget of $20C$ across all 6 datasets.

Table 2 compares the mean fidelity and its variance between the original CEGA and the three ablation models in the 6 datasets of interest, while Figure 3 shows a similar pattern for accuracy and F1 score is available in Appendix B.4. We summarize the key observations below: (1) From the perspective of model performance, CEGA demonstrates comparable to significantly better average performance than ablated models across different test datasets and metrics, particularly outperforming models where centrality or uncertainty is ablated by a large margin. This highlights the pivotal rule of these two components in identifying informative nodes for querying in early cycles. (2) From the perspective of consistency of estimates, CEGA provides more stable estimates across all metrics compared to the model with diversity ablated, especially for the Amazon-Computer and DBLP datasets. This reveals that the diversity component of CEGA plays a crucial role in later querying cycles by supporting performance stability.

## 5. Related Work

**Query-Efficient Model Extraction Attack.** (Tramèr et al., 2016) pioneered the study of MEA with high-fidelity extraction towards target black-box MLaaS models. (Pal et al., 2020) applies active learning techniques that dynamically adjust query selection based on self-feedback to extract deep classifiers in the domain of image and text, showing that querying 10% to 30% of samples from the dataset can yield a high-fidelity extraction model. Recent work shows that budget-sensitive MEA is feasible even for the data-free setting, where the attacks are performed without any in-distribution data (Lin et al., 2023b). Another proposed way to obtain a better query efficiency in MEA is to simultaneously train two clone models with the same samples and force them to learn from mismatching samples (Rosenthal et al., 2023). (Dai et al., 2023) employs clustering-based data reduction to minimize information redundancy in the query pool and realize query efficiency for the task of MEA in NLP. However, all these works on query-efficient MEA fail to consider graph-based model extraction.

**Model Extraction Attack in Graph Learning.** (Oliynyk et al., 2023) systematizes MEA on multiple model types, summarizes defense strategies based on available resources, and highlights an upward trend of popularity of literature on the attacks towards and defense for ML models. Recently, the research interest has pivoted towards the application of MEA on graph-related models. (DeFazio & Ramesh, 2019) first consider an adversarial model extraction approach for a graph-structured dataset. (Wu et al., 2022) provides a comprehensive analysis of GNN-based MEAs under various categories based on the attacker's knowledge of target graph structure and node attributes. (Shen et al., 2022) claims a significant contribution by proposing an inductive model structure that allows the attack graph to add new nodes to the existing model without the necessity of retraining. This development overcomes a major limitation of the prevalent GCN-based transductive approach (Kipf & Welling, 2017) that requires model retraining with the introduction of a new attacking or testing node. Generally, the GNN-based MEAs can be divided into two primary categories, distinguished by the attacker's knowledge of the target model's graph structure (Oliynyk et al., 2023). Go beyond existing publications, our work addresses a serious concern regarding the efficiency and budget limitation in model extraction on graph learning by subsequently improving the practicality.

## 6. Conclusion

In this paper, we introduce CEGA, a cost-effective framework specialized in node querying for graph-based model extraction. In particular, we formulate and study the problem of budget-constrained model extraction on graphs, where the objective is to maximize the extracted model's performance and resemblance to the target model with a minimized query budget. To overcome this challenge, we develop CEGA by designing an adaptive node selection strategy that effectively queries the most informative nodes based on incremental history information accumulated in the training progress. We present a theoretical guarantee on the feasibility and efficiency of our approach in measuring the uncertainty of history-based interim predictions for candidate nodes. Extensive experiments on real-world graph datasets demonstrate CEGA's superiority over state-of-the-art baselines across multiple key aspects. Looking ahead, two future directions warrant further exploration. First, our current framework is based on a *transductive* assumption, and we aim to extend the CEGA framework to *inductive* GNNs, following previous investigation by (Shen et al., 2022). Second, as suggested by (Guan et al., 2024), refining our approach by leveraging edge information, especially in the early query cycles when the number of selected nodes is small, could further improve CEGA's performance.

## Acknowledgements

Y.D. acknowledges funding in part by Start-Up Grant and FYAP (First Year Assistant Professor) Grant from Florida State University, Tallahassee, FL, USA.

## Impact Statement

We introduce CEGA (Cost-Efficient Graph Acquisition), a framework to deploy model extraction and acquisition on Graph Neural Networks (GNNs) under realistic constraints of limited query budgets and structural complexity.

For practitioners in high-stakes fields, CEGA formalizes the problem of *GNN Model Extraction With Limited Budgets*, laying a foundation for the development of practical defenses against GNN-based model extraction attacks (MEAs) against Machine Learning as a Service (MLaaS).

For researchers, CEGA reveals its non-adversarial potential of GNN extraction and acquisition in domains where expert labeling is prohibitively expensive and large-scale training is impractical. Ethical model acquisition offers a viable path to democratize high-performance GNNs and adapt them to specialized downstream tasks.

We emphasize the responsible use of CEGA, as its insights should be used to strengthen MLaaS security and advocate ethical research under limited resources.

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

# A. Proofs to Theoretical Analysis

In this section, we provide the proof to the theoretical contribution of CEGA.

*Proof to Proposition 3.1:* Before we calculate the complexity of the $\gamma$th cycle of CEGA, we need to conduct the forward step of $f_{\gamma-1}$ to obtain the softmax scores for each nodes of interest that are used in the subsequent procedures of CEGA. (Blakely et al., 2021) shows that the forward step for an $L$-layer GCN has time complexity $O(LN^2d + LNd^2)$ and space complexity $O(N^2 + Ld^2 + LNd)$. The output takes $O(CN + Nh)$ space and is shared among the following tasks. Here, $h$ is the dimension of the embeddings.

For CEGA's entropy-based approach to evaluate the uncertainty based on historical information, the space is required for $O(Cn_{\gamma-1})$ softmax scores. Our task is to calculate the entropy of the $O(n_{\gamma-1})$ nodes, which involves the computation of $O(Cn_{\gamma-1})$. Sorting the nodes involves $O(n_{\gamma-1} \log n_{\gamma-1})$ time complexity and $O(n_{\gamma-1})$ space usage.

For the more resource-consuming perturbation-based alternative, we first consider the complexity involved each time we redo the perturbation. For each time we take the perturbation, we prepare the perturbed attributes for the nodes in $\mathcal{G}_a$, which takes $O(Nd)$ space and has $O(Nd)$ time consumption. As we pass the perturbed attributes forward through the GNN model and calculate the softmax scores for the perturbed scores, the time complexity is $O(LN^2d + LNd^2)$. For each time we redo the perturbation, the output takes $O(CN)$ space, and a time complexity of $O(Cn_{\gamma-1})$ is required to compare the labels derived from the perturbed softmax score and the original score. The output is stored in a vector with $O(n_{\gamma-1})$ dimensions, where the space for temporarily perturbed attributes and softmax scores can be released after each perturbation. We redo the perturbation procedure $S$ times and sort the nodes based on $\mathcal{L}_2^\gamma$ with $O(n_{\gamma-1} \log n_{\gamma-1})$ time complexity.

Taking summations on all the procedures implies that the additional time complexity is $O(Cn_{\gamma-1} + n_{\gamma-1} \log n_{\gamma-1})$ for CEGA's entropy-based approach. For the perturbation-based alternative, the additional time complexity is $O(SNd + SLN^2d + SLNd^2 + SCn_{\gamma-1} + n_{\gamma-1} \log n_{\gamma-1})$. Given all the assumptions of Proposition 3.1, we summarize that the time complexity are $O(CN + N \log N)$ and $O(SLN^2d + SLNd^2)$, respectively. Under a similar step of calculation, we have that the space complexity of CEGA's entropy-based approach and the perturbation-based alternative are $O(CN)$ and $O(N^2 + Ld^2 + LNd)$, respectively.

*Proof to Theorem 3.2:* In the proof, we consider a generic GNN network. Taking the GCN model (Kipf & Welling, 2017) as an example, we have that

$$\mathbf{G}^{(1)} = \sigma\big(f(\mathbf{A})\mathbf{X}\mathbf{W}^{(1)}\big); \quad \mathbf{G}^{(\ell+1)} = \sigma\big(f(\mathbf{A})\mathbf{G}^{(\ell)}\mathbf{W}^{(\ell+1)}\big).$$

Here $f(\mathbf{A}) = \widehat{\mathbf{D}}^{-1/2}\widehat{\mathbf{A}}\widehat{\mathbf{D}}^{-1/2}$. The last layer before the output is conducted by the softmax procedure. For one specific node $\tau$, we consider a generic two-layer GNN model

$$\mathbf{g}_\tau^{(1)} = \sigma\Big(\mathbf{W}_{sa}^{(1)}\mathbf{x}_\tau + \mathbf{W}_n^{(1)} \sum_{\mu \in \mathcal{N}_\tau} \mathbf{x}_\mu\Big); \quad \mathbf{g}_\tau^{(2)} = \mathbf{W}_{out}\mathbf{g}_\tau^{(1)} + \mathbf{b}; \quad \widehat{\mathbf{y}}_\tau = \mathrm{softmax}(\mathbf{g}_\tau^{(2)}). \tag{5}$$

Here $\mathbf{W}_{sa}^{(1)}$ and $\mathbf{W}_n^{(1)}$ represents the weights assigned to the self-attention term and the neighborhood for the node $\tau$, specifically. $\mathcal{N}_\tau$ represents the neighborhood of the node $\tau$. Substituting the node attributes $\mathbf{x}_\tau$ to $\widetilde{\mathbf{x}}_\tau$ indicates that

$$\widetilde{\mathbf{g}}_\tau^{(1)} = \sigma\Big(\mathbf{W}_{sa}^{(1)}\widetilde{\mathbf{x}}_\tau + \mathbf{W}_n^{(1)} \sum_{\mu \in \mathcal{N}_\tau} \widetilde{\mathbf{x}}_\mu\Big); \quad \widetilde{\mathbf{g}}_\tau^{(2)} = \mathbf{W}_{out}\widetilde{\mathbf{g}}_\tau^{(1)} + \mathbf{b}; \quad \widehat{\mathbf{y}}_\tau^p = \mathrm{softmax}(\widetilde{\mathbf{g}}_\tau^{(2)}). \tag{6}$$

Aggregating (5) and (6) indicates that

$$\big\|\widehat{\mathbf{y}}_\tau - \widehat{\mathbf{y}}_\tau^p\big\|_2 \leq \mathcal{C}_{soft}\big\|\mathbf{g}_\tau^{(2)} - \widetilde{\mathbf{g}}_\tau^{(2)}\big\|_2 \leq \mathcal{C}_{soft}\big\|\mathbf{W}_{out}\big\|_2\big\|\mathbf{g}_\tau^{(1)} - \widetilde{\mathbf{g}}_\tau^{(1)}\big\|_2$$

$$\leq \mathcal{C}_{soft}\,\mathcal{C}_\sigma\big\|\mathbf{W}_{out}\big\|_2\big\|\mathbf{W}_{sa}^{(1)}\big\|_2\big\|\mathbf{x}_\tau - \widetilde{\mathbf{x}}_\tau\big\|_2 + \mathcal{C}_{soft}\,\mathcal{C}_\sigma\big\|\mathbf{W}_{out}\big\|_2\big\|\mathbf{W}_n^{(1)}\big\|_2\Big\|\sum_{\mu \in \mathcal{N}_\tau}\big(\mathbf{x}_\mu - \widetilde{\mathbf{x}}_\mu\big)\Big\|_2.$$

Here $\mathcal{C}_{soft}$ and $\mathcal{C}_\sigma$ denotes the Lipschitz constant for softmax function and the activation function $\sigma(\cdot)$, respectively. The norms $\big\|\mathbf{W}_{out}\big\|_2$, $\big\|\mathbf{W}_{sa}^{(1)}\big\|_2$, and $\big\|\mathbf{W}_n^{(1)}\big\|_2$ are bounded from above given that the estimation function is bounded after one step of model fitting. For simplicity, we re-arrange the terms and form the inequality such that

$$\big\|\widehat{\mathbf{y}}_\tau - \widehat{\mathbf{y}}_\tau^p\big\|_2 \leq \eta_1\big\|\mathbf{x}_\tau - \widetilde{\mathbf{x}}_\tau\big\|_2 + \eta_2\Big\|\sum_{\mu \in \mathcal{N}_\tau}\big(\mathbf{x}_\mu - \widetilde{\mathbf{x}}_\mu\big)\Big\|_2,$$

for some positive constants $\eta_1, \eta_2 < \infty$. Given that $\mathbf{x}_\mu - \widetilde{\mathbf{x}}_\mu \sim \mathcal{N}(0, \epsilon^2)$ for any $\mu \in \mathcal{N}_\tau$, we can apply Hoeffding's inequality, which implies that

$$\mathbb{P}\left( \sum_{\mu \in \mathcal{N}_\tau} \left( \mathbf{x}_\mu - \widetilde{\mathbf{x}}_\mu \right) \geq t \right) \leq \exp\left( -\frac{t^2}{2 \, |\mathcal{N}_\tau| \epsilon^2} \right).$$

We then select

$$\epsilon = \min\left\{ \frac{\zeta}{\eta_1 \sqrt{2 \log(1/\delta)}}, \frac{\zeta}{\eta_2 \sqrt{2|\mathcal{N}_\tau| \log(1/\delta)}} \right\},$$

where we guarantee that the difference of the outcome label probability has an upper bound with a large probability.

## B. Supplementary Results and Discussion

In this section, we elaborate the discussion of the additional results of the experiment based on our implementation of CEGA, which is available at `https://github.com/LabRAI/CEGA`, to a series of widely studied benchmark graph datasets.

### B.1. Datasets

| Dataset | #Nodes | #Edges | #Features | #Classes |
|---|---|---|---|---|
| AmzComputer | 13,752 | 491,722 | 767 | 10 |
| AmzPhoto | 7,650 | 238,163 | 745 | 8 |
| CoauthorCS | 18,333 | 163,788 | 6,805 | 15 |
| CoauthorPhysics | 34,493 | 495,924 | 8,415 | 5 |
| Cora Full | 19,793 | 126,842 | 8,710 | 70 |
| DBLP | 17,716 | 105,734 | 1,639 | 5 |

*Table 3.* Dataset statistics

Table 3 presents the statistics of six benchmark datasets used in our study, covering a range of node, edge, feature, and class distributions. Amazon-Computer and Amazon-Photo are e-commerce co-purchase networks characterized by dense connectivity. Coauthor-CS and Coauthor-Physics represent academic collaboration graphs with a larger number of features. Cora_Full and DBLP are citation network datasets where nodes represent academic papers and edges denote citation relationships. Cora Full spans diverse machine learning subfields with 70 classes, while DBLP focuses on computer science publications with five broad research categories. These datasets provide diverse graph structures and feature distributions for evaluating model performance.

### B.2. Setup of Hyperparameters

**Setup of GNN Model Extraction**    We follow the *Attack 0* framework of (Wu et al., 2022) to perform GNN model extraction. Initially, we train a target model, $f_T$, for 1000 epochs with a learning rate of 1e-3, which provides predictions for surrogate model training. If training and test sets are not provided, we randomly select 60% of the nodes for training and use the remaining 40% for testing. This serves as the initial setup; however, following the *Attack 0* framework, these masks are later adjusted based on whether the nodes are subject to queries for extraction.

**Setup of Node Selection Models**    For our experiments, we randomly set the candidate node pool $\mathcal{V}_a$ comprising 10% of the nodes in graphs with fewer classes, including Amazon-Computer, Amazon-Photo, Coauthor-CS, Coauthor-Physics, and DBLP. For graph with significantly higher number of classes (e.g., Cora-Full, which has 70 classes), the pool includes 25% of the nodes. Our setup is inspired by widely accepted works, such as (Wu et al., 2022; Shen et al., 2022). In the initialization step, we randomly select 2 nodes from each class across all the tested datasets, resulting in a total of $2C$ nodes, where $C$ is the number of classes. In practice, this procedure remains feasible as the extractors can attain partial knowledge of the class distribution through domain expertise or external sources, especially when such knowledge offers strategic advantages in building a high-fidelity extracted model. For the remaining budget, we employ different node selection methods, with the total budget capped at $20C$.

For the baseline node selection methods, hyperparameters are set as follows. In GRAIN (Ball-D), the radius $r$ is fixed at 0.005 for all datasets, while in GRAIN (NN-D), $\gamma$ is set to 1. For AGE, we adopt the time-sensitive parameter setting, where $\gamma_t \sim \text{Beta}(1, n_t)$, with $n_t$ increasing as iterations progress, defined as $n_t = 1.05 - 0.95^t$. Here, $t$ denotes the number of iterations. The parameters $\alpha_t$ and $\beta_t$ are set as $\alpha_t = \beta_t = \frac{1-\gamma_t}{2}$.

For our proposed method, in cycle $\gamma$, CEGA queries $\kappa = 1$ node and trains a 2-layer GCN model with $\{\mathcal{V}_\gamma, \mathcal{G}_a\}$ for $E = 1$ epoch. In the analysis for node diversity, we set the weight $\rho = 0.8$ to ensure that the order $\mathcal{R}_3^\gamma$ is designed to prioritize the nodes associated with underrepresented prediction labels. For the weighted average ranking mechanism, we set

$$\omega_1(\gamma) = \alpha_1 + \Delta e^{-\lambda\gamma}; \ \ \omega_2(\gamma) = \alpha_2 + \Delta\big(1 - e^{-\lambda\gamma}\big); \ \ \omega_3(\gamma) = \alpha_3(1 - e^{-\gamma}). \tag{7}$$

We design this weighting approach under the heuristics such that the subgraph structure is the most important information when the information gathered in the history is not accurate enough to guide further queries. As the number of queries increases, the contribution from history information becomes more prominent, and diversity concerns need to be considered more seriously. In practice, we set the initial weight coefficients as $\alpha_1 = \alpha_2 = \alpha_3 = 0.2$, the measurement of the initial weight gap between $\mathcal{R}_1^\gamma$ and $\mathcal{R}_2^\gamma$ as $\Delta = 0.6$, the measurement of the curvature for the weight changes as $\lambda = 0.3$. The CEGA hyperparameters $(\alpha_1, \alpha_2, \alpha_3, \Delta, \lambda)$ are applied uniformly across all the tested graph datasets to mitigate potential concerns of tuning bias.

After the node selection process, we train a 2-layer GCN with a hidden dimension of 16. The model is optimized with a learning rate of 1e-3 and trained for 1000 epochs. For AGE, we apply a warm-up period of 400 epochs. All experiments are conducted on two NVIDIA RTX 6000 Ada GPUs. Model performance is evaluated for node selections ranging from $2C$ to $20C$, with evaluations performed at every $C$. Selected nodes are trained for 1000 epochs using a learning rate of 1e-3.

## B.3. Model Performance Gap

*Table 4.* Performance gaps between budget-constrained models and subgraph models, measured by Accuracy, Fidelity, and F1, across various datasets. The best results are in **bold**.

|          |               | CoCS | CoP | AmzC | AmzP | Cora_Full | DBLP |
|----------|---------------|------|-----|------|------|-----------|------|
| Accuracy | **Random**        | $2.68 \pm 0.6$ | $3.47 \pm 1.0$ | $3.60 \pm 1.1$ | $2.07 \pm 1.6$ | $1.71 \pm 0.5$ | $12.24 \pm 1.3$ |
|          | **GRAIN(NN-D)**   | $1.69 \pm 0.6$ | $1.87 \pm 0.8$ | $3.41 \pm 1.1$ | $1.56 \pm 0.4$ | $-0.09 \pm 1.1$ | $13.14 \pm 1.0$ |
|          | **GRAIN(ball-D)** | $2.02 \pm 0.6$ | $1.93 \pm 1.0$ | $4.78 \pm 1.3$ | $2.58 \pm 1.2$ | $0.04 \pm 1.0$ | $12.98 \pm 1.2$ |
|          | **AGE**           | $\mathbf{0.78 \pm 0.4}$ | $1.56 \pm 0.3$ | $2.33 \pm 0.9$ | $1.39 \pm 1.8$ | $0.90 \pm 0.4$ | $8.98 \pm 2.0$ |
|          | **CEGA**          | $0.91 \pm 0.4$ | $\mathbf{1.39 \pm 0.4}$ | $\mathbf{1.19 \pm 0.8}$ | $\mathbf{0.58 \pm 0.3}$ | $\mathbf{-1.36 \pm 0.2}$ | $\mathbf{7.79 \pm 1.1}$ |
| Fidelity | **Random**        | $3.20 \pm 0.7$ | $4.20 \pm 1.0$ | $4.24 \pm 1.2$ | $2.34 \pm 1.6$ | $3.84 \pm 0.5$ | $14.85 \pm 1.7$ |
|          | **GRAIN(NN-D)**   | $2.26 \pm 0.7$ | $2.49 \pm 1.0$ | $4.01 \pm 1.4$ | $1.74 \pm 0.6$ | $1.34 \pm 1.7$ | $16.11 \pm 1.2$ |
|          | **GRAIN(ball-D)** | $2.61 \pm 0.7$ | $2.50 \pm 1.2$ | $5.77 \pm 1.7$ | $3.06 \pm 1.4$ | $1.68 \pm 1.2$ | $15.52 \pm 1.4$ |
|          | **AGE**           | $\mathbf{1.02 \pm 0.4}$ | $2.07 \pm 0.4$ | $2.70 \pm 1.0$ | $1.77 \pm 2.4$ | $2.31 \pm 0.7$ | $11.50 \pm 2.2$ |
|          | **CEGA**          | $1.25 \pm 0.5$ | $\mathbf{1.84 \pm 0.5}$ | $\mathbf{1.69 \pm 0.8}$ | $\mathbf{0.56 \pm 0.5}$ | $\mathbf{0.06 \pm 0.4}$ | $\mathbf{9.98 \pm 1.2}$ |
| F1       | **Random**        | $6.72 \pm 1.6$ | $5.44 \pm 1.8$ | $4.61 \pm 2.8$ | $3.69 \pm 3.5$ | $0.49 \pm 0.3$ | $19.78 \pm 5.4$ |
|          | **GRAIN(NN-D)**   | $2.77 \pm 1.5$ | $2.69 \pm 1.1$ | $4.00 \pm 1.2$ | $2.87 \pm 0.8$ | $-0.97 \pm 1.0$ | $19.10 \pm 4.0$ |
|          | **GRAIN(ball-D)** | $3.04 \pm 1.3$ | $2.75 \pm 1.5$ | $8.53 \pm 5.3$ | $3.85 \pm 1.7$ | $-0.93 \pm 0.3$ | $16.95 \pm 3.7$ |
|          | **AGE**           | $\mathbf{0.84 \pm 0.6}$ | $2.13 \pm 0.5$ | $4.57 \pm 5.1$ | $1.30 \pm 1.6$ | $-1.12 \pm 0.3$ | $11.93 \pm 2.9$ |
|          | **CEGA**          | $1.09 \pm 0.7$ | $\mathbf{1.89 \pm 0.6}$ | $\mathbf{0.86 \pm 2.8}$ | $\mathbf{0.80 \pm 0.7}$ | $\mathbf{-3.43 \pm 0.5}$ | $\mathbf{10.02 \pm 1.0}$ |

Table 4 quantifies the performance gap between budget-constrained models and subgraph models across various datasets, using Accuracy, Fidelity, and F1 as evaluation metrics. A smaller gap indicates a more effective node selection strategy, with negative values suggesting cases where the budget-constrained model outperforms the subgraph model. Notably,

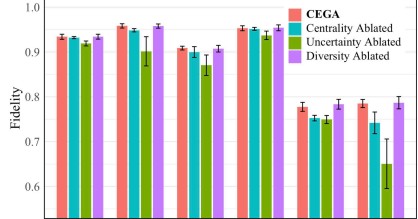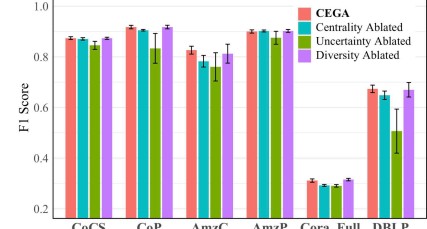

*Figure 3.* Ablation study results on accuracy (left), fidelity (middle), and F1 score (right) for CEGA and variants with one of the three node evaluation modules removed.

CEGA almost outperforms the benchmark models across all three metrics, demonstrating its effectiveness in maintaining model performance under stringent budget constraints. Additionally, this table provides detailed numerical insights that complement the trends illustrated in Figure 2.

### B.4. Ablation Study

To address **RQ3**, an ablation study is conducted where we implement two of the three analyses proposed in Section 3.2. Specifically, we compare the original CEGA model against three variants: (1) *CEGA with Centrality Module Ablated*: A variant removing the centrality-based selection mechanism, which we expect to evaluate the contribution of the subgraph model structure to the selection of nodes to be queried; (2) *CEGA with Uncertainty Module Ablated*: A variant removing the contribution of prediction uncertainty under the guidance of history information, which we expect to evaluate the contribution of history information extracted from previous queries; (3) *CEGA with Diversity Module Ablated*: A variant removing the contribution that enhances the diversity of the selected nodes, which we expect to evaluate the contribution of node diversity in providing a more stable estimation with a smaller variation across different random initialization setups. The setup of our ablation study follows the standard of the most recent works on GNN node classification tasks, such as (Lin et al., 2025). In practice, we set the respective cycle-specific weight $\omega_i(\gamma) = 0$, as specified in (7), among all $\gamma \in \{1, 2, ..., \Gamma\}$ for the specific index $i \in \{1, 2, 3\}$.

The results of the ablation study, as shown in Figure 3, are consistent across all three performance metrics, namely accuracy, fidelity, and F1 score. This indicates that the ablation study yields stable findings regardless of the evaluation criterion. This alignment reinforces the robustness of our proposed approach and suggests that each module contributes meaningfully to the overall performance of the model.

