# OpenReview forum: "CEGA: A Cost-Effective Approach for Graph-Based Model Extraction and Acquisition"
_ICML.cc/2025/Conference — ICML 2025 poster_

### Official Review · Reviewer_JNMU · 2025-03-13

**Overall Recommendation:** 3

**Summary:**

This paper introduces CEGA for model extraction attacks against GNNs with limited query budgets. The authors address practical constraints in real-world attack scenarios by designing a node querying strategy that incrementally refines node selection over multiple learning cycles. CEGA integrates three key considerations: representativeness (selecting nodes that capture graph structure), uncertainty (prioritizing nodes near decision boundaries), and diversity (ensuring comprehensive exploration). Experiments on six benchmark datasets demonstrate that CEGA outperforms baseline approaches in terms of accuracy, fidelity, and F1 score under strict query-size constraints. The authors provide theoretical guarantees for their approach and highlight the vulnerability of GNNs to extraction attacks despite resource limitations.

**Claims And Evidence:**

The paper's primary claims regarding CEGA's superiority over baseline methods are generally well-supported by comprehensive experimental evidence. The authors demonstrate this through extensive comparisons on six different benchmark datasets, showing consistently better performance for CEGA across accuracy, fidelity, and F1 score metrics. The performance gap analysis between budget-constrained and full subgraph models further strengthens their claims.

However, the claim that CEGA achieves "cost-effective" attacks would be stronger with a direct efficiency comparison (e.g., computational cost, time complexity in practice) against baseline methods, beyond just the theoretical complexity analysis provided.

**Essential References Not Discussed:**

No

**Experimental Designs Or Analyses:**

The experimental design is generally robust. However, I identified several limitations:

1. The attack node pool is restricted to 10% of nodes (25% for Cora-Full), which may not reflect real-world knowledge constraints.

2. Hyperparameter selection process isn't fully detailed, raising questions about potential tuning bias.

**Methods And Evaluation Criteria:**

The proposed methodology is logically sound and well-suited for the problem. The three-component node selection strategy addresses key challenges in model extraction with limited budgets. The evaluation criteria appropriately measure both the performance of the extracted model and its similarity to the target model.

The benchmark datasets represent diverse graph structures and applications, providing a robust testbed. However, the evaluation would be stronger with more real-world MLaaS settings where API rate limits and costs are explicitly modeled, rather than simply using node count as a proxy for query budget.

**Other Comments Or Suggestions:**

see above

**Other Strengths And Weaknesses:**

Other weaknesses:

1. The initial query selection (2 nodes per class) assumes prior knowledge of class distribution, which may be unrealistic in practical attack scenarios.

2. The ablation study removes one component at a time, but doesn't explore interactions between components (e.g., combining only representativeness and uncertainty)

**Questions For Authors:**

see above

**Relation To Broader Scientific Literature:**

No

**Theoretical Claims:**

I reviewed the two main theoretical claims (Theorems 3.1 and 3.2) and their proofs in Appendix B. Theorem 3.1 on complexity evaluation appears correct, leveraging established complexity results for GCN operations. The proof appropriately breaks down computational components and aggregates them to derive the overall complexity. Theorem 3.2 on the existence of a feasible permutation parameter ε is also sound. The proof uses Hoeffding's inequality to establish probabilistic bounds on the perturbation's effect, ensuring stability in the model's predictions. Both proofs support the practical feasibility of the proposed approach, especially for the uncertainty evaluation component.

---

> ### Author Rebuttal · Authors · 2025-04-01
>
> Thank you for your valuable feedback. We provide a point-to-point reply below.
>
> **Response to Review on Claims**: Thank you for pointing this out. We would like to clarify that the *cost-effectiveness* we attribute to CEGA refers to two major aspects. First, CEGA shows its ability to **achieve high fidelity to $f_{\mathrm{T}}$ using fewer queried nodes**, which we show in **Figure 1**. In pay-per-query settings, this translates into considerable cost savings for those who aim to replicate an expensive target model under a limited budget. Second, CEGA shows computational efficiency as being **1.5\~4x more efficient** in terms of running time across most tested cases and datasets than AGE, which delivers competitive performance with CEGA on **CoCS** and **CoP**. This observation aligns with CEGA's theoretical efficiency, as we presented in **Section 3.3**.
>
> **Response to Review on Evaluation**: We thank you for your constructive feedback. First, we agree on the importance of assessing CEGA's cost-effectiveness in a simulated real-world setting involving modeled constraints. Our current evaluation setup for CEGA is adaptable for such a setting, as an evaluation on a different number of nodes selected by CEGA or benchmark models can be assessed by modifying the right cutoff for the trajectories in **Figure 1**. We will thoroughly evaluate CEGA's performance and time complexity under simulated real-world MLaaS settings in the next version of our manuscript.
>
> **Response to Experimental Design \#1**: We thank you for your thoughtful feedback. To maintain consistency with common practice in the field, we follow the setup of numerous widely accepted works on MEAs against GNNs, notably [1] and [2]. Heuristically, we want to show CEGA's effectiveness in selecting informative nodes with **partial knowledge** to the target graph that contains **sufficient information** for MEA tasks. We use 25% for Cora-Full due to its significantly higher number of classes. We will provide detailed justification for our MEA framework setup and conduct additional experiments under varying attack node pool sizes in our next version.
>
> [1] Model Extraction Attacks on Graph Neural Networks: Taxonomy and Realisation, Wu et al., ASIA CCS 2022.
>
> [2] Model Stealing Attacks against Inductive Graph Neural Networks, Shen et al., IEEE Symposium on Security and Privacy, 2022.
>
> **Response to Experimental Design \#2**: We thank you for pointing this out. In our practice, we follow the heuristics outlined in **Section 3.2** and design a setup as specified in **Equation (7)**. Specifically, we use **grid search**, which is commonly used in high-impact work on GNN evaluation [1], to find the best combination of hyperparameters for CEGA, namely initial weights $\alpha_1$, $\alpha_2$, $\alpha_3$, weight gap $\Delta$, and weight change curvature $\lambda$, under this setup. To ensure generalizability, we apply the same set of hyperparameters across all 6 datasets. We will include further details on our tuning process in our next version.
>
> [1] Pitfalls of Graph Neural Network Evaluation, Shchur et al., arXiv:1811.05868, 2018.
>
> **Response to Other Weakness \#1**: We thank you for raising this point. We would like to clarify that ensuring even class distribution for the initial query is a widely accepted common practice in the field, originating from fundamental contributions to GNN research [1] and continuously adopted by subsequent works we use as benchmarks [2][3]. In practice, it is reasonable to assume that the adversaries can attain partial knowledge of the class distribution through domain expertise in fields like *Drug-Target Interaction* or external sources such as *accounts outside the target network*. They will focus on obtaining such knowledge if they believe it brings them long-term monetary or strategic advantages based on a high-fidelity extracted model.
>
> [1] Revisiting Semi-supervised Learning with Graph Embeddings, Yang et al., ICML 2016.
>
> [2] Active Learning for Graph Embedding, Cai et al., arXiv:1705.05085, 2017.
>
> [3] GRAIN: Improving Data Efficiency of Graph Neural Networks via Diversified Influence Maximization, Zhang et al., VLDB 2021.
>
> **Response to Other Weakness \#2**: We thank you for pointing this out. Our ablation study setup stated in **Section 4.4** that removes the contribution of one out of the three modules, namely *representativeness*, *uncertainty*, and *diversity*, follows the settings of the most recent works on GNN node classification tasks [1]. In conclusion,
>
> 1. CEGA outperforms nearly all the ablated models in most datasets, specifically models with only (1) a combination of only representativeness and diversity and (2) a combination of only uncertainty and diversity.
> 2. CEGA provides more stable estimates across most datasets and metrics compared to the model that combines only representativeness and uncertainty.
>
> [1] Semantic GNN with Multi-Measure Learning for Semi-Supervised Classification, Lin et al., EAAI 2025.

---

### Official Review · Reviewer_oxao · 2025-03-14

**Overall Recommendation:** 3

**Summary:**

The paper proposes a method for model extraction attack, in the setting where the number of prediction queries is extremely tight.
Node predictions are queries in different rounds and based on three criteria: representativeness,  uncertainty, and diversity.
For the entropy-based approach, time and space complexities are computed. The experiments show that the method is effective in marginally improving the metrics.

**Claims And Evidence:**

The improvements are often marginal, and sometimes significant.
Not included in score: In Tab. 1, in the 1st column the proposed method's metric is mistakenly in bold font, please correct it.

**Essential References Not Discussed:**

-

**Experimental Designs Or Analyses:**

The approach is empirically validated.

**Methods And Evaluation Criteria:**

I have questions and doubts about  the discussion of Sec. "History-Based Analysis for Uncertainty".
The variable $\tao$ is generated from a normal distribution, which are added to the node attributes. Why doing so is referred to as "permutation" in the manuscript?

In summary, although the paper shows that the 3 introduced criteria lead to marginal improvement, I'm not sure if the contribution is sufficient for an ICML paper. Moreover, most improvements are marginal.

**Other Comments Or Suggestions:**

-

**Other Strengths And Weaknesses:**

The writing and notation were ambiguous to me throughout the paper.

**Questions For Authors:**

-

**Relation To Broader Scientific Literature:**

-

**Theoretical Claims:**

The computation and space complexity of the entropy-based approach is computed and is presented as a theorem.
Although the time/space complexity is important, but in my opinion it shouldn't be presented as a theorem nor does it count as a contribution.

---

> ### Author Rebuttal · Authors · 2025-04-01
>
> Thank you for your valuable feedback. We provide a point-to-point reply below to address the mentioned questions and concerns.
>
> > **Reviewer**: The improvements are often marginal, and sometimes significant. I'm not sure if the contribution is sufficient for an ICML paper. Moreover, most improvements are marginal.
>
> **Authors**: We thank you for raising this concern. We highlight the primary technical contribution of CEGA as achieving a highly challenging task of achieving strong and consistent performance in terms of saving query nodes while achieving high-fidelity model extraction. Such an advantage is especially evident in **Amazon_Computer**, **Cora_Full**, and **DBLP**, as shown in the trajectories at **Figure 1**. In contrast, AGE and GRAIN both show limitations on instability, with AGE underperforming on **Amazon_Photo** and **Cora_Full**, and GRAIN struggling on **Coauthor_CS**, **Amazon_Computer**, and **DBLP**. As a remark, many published papers in related fields have not achieved significantly superior performance across all the tested datasets, such as [1][2][3], with some underperforming against benchmarks by a larger margin than how CEGA has performed. We believe that CEGA makes a meaningful contribution to the further advancement of our community by introducing a practical and underexplored problem supported by comprehensive and stable empirical results. We are confident that our work can guide future efforts in research question formulation, method development, and theoretical analysis in the area of GNN security and democratization.
>
> [1] Graph Attention Networks, Veličković et al., ICLR 2018.
>
> [2] GNN-FiLM: Graph Neural Networks with Feature-wise Linear Modulation, Brockschmidt, ICML 2020.
>
> [3] Directional Message Passing for Molecular Graphs, Gasteiger et al., ICLR 2020.
>
> > **Reviewer**: In Tab. 1, in the 1st column the proposed method's metric is mistakenly in bold font, please correct it.
>
> **Authors**: We appreciate your attention to detail. We acknowledge and appreciate your suggestion to apply bold fonts only to the best performance for each task and will synchronize the new standard in the next version of our manuscript. We hope that our commitment addresses your concerns about the clarity of our work.
>
> > **Reviewer**: I have questions and doubts about the discussion of Sec. "History-Based Analysis for Uncertainty". The variable
>  is generated from a normal distribution, which are added to the node attributes. Why doing so is referred to as "permutation" in the manuscript?
>
> **Authors**: We thank you for pointing this out. We agree that the term *permutation* is not precise enough. In the next version of our manuscript, we will replace this terminology with *perturbation*, which more accurately describes the application of a random Gaussian oscillation to the node attributes. This procedure allows us to evaluate the nodes' uncertainty regarding classification labels evaluated by the history-inspired interim model $f_{\mathrm{T}}$ under attribute variation. We hope our committed edits will address your concern about the clarity of our terminology.
>
> > **Reviewer**: Although the time/space complexity is important, but in my opinion it shouldn't be presented as a theorem nor does it count as a contribution.
>
> **Authors**: We appreciate your feedback and understand your concern about presenting CEGA's time and space complexity as a standalone theorem. In our next version, we will no longer treat the results of our computation on CEGA's time/space complexity as a theorem. We hope our committed edits will address your concern about the arrangement of contents covered in the manuscript.
>
> > **Reviewer**: The writing and notation were ambiguous to me throughout the paper.
>
> **Authors**: We thank you for the feedback regarding the writing and notation of our manuscript. We are eager to know about the specific locations that make you feel the ambiguity in our writing and notation hinders the understanding of our work. We are open to further discussion on notation consistency and the overall clarity of our current manuscript, and we hope that more specific feedback from you in this aspect will further improve the quality of our work.

---

> > ### Comment · Reviewer_oxao · 2025-04-07
> >
> > Thanks for the explanations. I increased my score accordingly.

---

### Official Review · Reviewer_E9DY · 2025-03-16

**Overall Recommendation:** 4

**Summary:**

This paper explores the vulnerability of Graph Neural Networks (GNNs) to Model Extraction Attacks (MEAs), particularly in scenarios with limited query budgets and initialization nodes. The authors propose a novel node querying strategy that incrementally refines the selection of nodes over multiple learning cycles, using historical information to enhance extraction efficiency.The paper makes three main contributions: it introduces a novel problem formulation for Model Extraction Attacks (MEAs) against GNNs with limited query budgets, focusing on node-level graph learning tasks; proposes the CEGA framework, which identifies the most beneficial queries during training by considering representativeness, uncertainty, and diversity; and provides extensive experiments on real-world datasets, demonstrating that CEGA outperforms existing methods in terms of fidelity, utility, and key performance metrics.

**Claims And Evidence:**

This article claims that although graph neural networks (GNNs) have demonstrated superior performance in many applications, they are vulnerable to model extraction attacks (MEAs) in machine learning as a service (MLaaS) environments.Attackers can steal the functionality of GNN models by strategically querying the target model, thereby replicating high fidelity models. Such attacks can lead to serious consequences, such as copyright and patent infringement, especially in the pharmaceutical industry where GNNs are widely used for predicting drug molecule target interactions. If these GNN models are extracted, it may threaten the trade secrets of pharmaceutical companies, leading to unauthorized redistribution and unfair competition, ultimately causing significant financial and reputational losses.

**Essential References Not Discussed:**

No

**Experimental Designs Or Analyses:**

The experiment used six common graph datasets (Coauthor CS, Amazon Computer, Cora Full, DBLP, etc.). These datasets cover various application scenarios, from collaborative networks to product recommendation systems, with broad representativeness.By comparing with benchmark methods such as Random, GRAIN (an active learning method based on neural networks), and AGE (another active learning method), the superiority of the CEGA method has been verified. The article evaluated the performance of various methods under different query constraints by testing different query budgets (from 2C to 20C nodes).The article also conducted ablation experiments to analyze the contribution of each module to overall performance by removing certain evaluation modules of CEGA, such as structural centrality, uncertainty, and diversity.

**Methods And Evaluation Criteria:**

The article used six commonly used graph datasets (such as Coauthor CS, Amazon Computer, Cora Full, DBLP, etc.), which cover various application scenarios from collaborative networks, product recommendations to academic citations. Tested on different datasets, CEGA demonstrated strong adaptability and robustness in various situations.The node selection strategy proposed by CEGA considers the representativeness, uncertainty, and diversity of nodes, thereby avoiding excessive concentration on certain nodes。

**Other Comments Or Suggestions:**

No

**Other Strengths And Weaknesses:**

originality：The paper proposes the CEGA (Cost Efficient Graph Attack) framework, which combines budget constraints and structural complexity, providing new ideas for efficient and low-cost model extraction in practical applications. This method has high originality in GNN security research.In addition, the paper innovatively improves attack efficiency by introducing multiple node selection criteria (representativeness, uncertainty, and diversity).
significance：The paper provides a meaningful framework for conducting GNN model extraction attacks on service (MLaaS) platforms, which can help protect systems using GNN in high-risk areas such as drug discovery, financial fraud detection, and healthcare from the threat of model leakage.
clarity：The structure of the paper is clear, first introducing the potential threats and challenges of GNN in practical applications, and then elaborating on the design principles, methods, and experimental evaluation of the CEGA framework in detail. The experimental part fully validated the effectiveness of the method and analyzed the contribution of each module through ablation experiments, ensuring the verifiability and reliability of the method.

**Questions For Authors:**

No

**Relation To Broader Scientific Literature:**

One of the biggest innovations of the article is the proposal of the budget constrained MEA problem. Most existing research has overlooked budget constraints in practical applications, while CEGA proposes a more practical attack method by introducing limitations on the number of queries and query nodes per round.Current research mainly focuses on the overall information or local structure of the graph, while CEGA introduces a comprehensive consideration of historical information and structural centrality, making attacks more efficient on actual complex graph structures.

**Theoretical Claims:**

Theorem 3.1 analyzes the temporal and spatial complexity of the CEGA method. The theorem states that the entropy calculation of the CEGA method introduces a time complexity of O (CN+N log N) and a space complexity of O (CN), where C is the number of categories and N is the number of nodes in the graph. This is based on the node selection and graph propagation process in the CEGA framework, and derives the time and space complexity.
Theorem 3.2 raises the issue of the existence of disturbance intensity ϵ. Specifically, when measuring the uncertainty of candidate nodes, there exists an appropriate disturbance intensity ϵ that ensures the stability of the model is not compromised after disturbance. This means that the CEGA method can ensure the effectiveness of the model when considering node disturbances, while not introducing excessive errors.

---

> ### Author Rebuttal · Authors · 2025-04-01
>
> Thank you for your comprehensive and thoughtful feedback towards the CEGA work. We sincerely appreciate the time and expertise you dedicated to carefully reviewing our manuscript. We would like to provide some further insights to our paper regarding your comments to fully address the potential concerns.
>
> > **Reviewer**: The node selection strategy proposed by CEGA considers the representativeness, uncertainty, and diversity of nodes, thereby avoiding excessive concentration on certain nodes.
>
> **Authors**: We thank you for highlighting the key understanding of CEGA. The major motivation for us to impose diversity in CEGA's node selection strategy is precisely to prevent over-concentration on particular subsets of the graph, which often consist of nodes sharing the same label. Avoiding such concentration patterns not only improves the fidelity of extracted model on classifying nodes belonging to underrepresented label categories, but also enhances the stability of CEGA across different initialization and expected knowledge from the adversaries side.
>
> > **Reviewer**: Theorem 3.1 analyzes the temporal and spatial complexity of the CEGA method.
>
> **Authors**: We appreciate your correct understanding of the theoretical complexity analysis. As raised by Reviewers oxao and JNMU, we acknowledge the necessity to make a clearer evaluation of both the theoretical and practical computation complexity of CEGA. In the next version of our manuscript, we will reorganize the presentation of Theorem 3.1 and related discussion in real-world MLaaS setup. We are open to further suggestions with you on refining the study of the computation complexity of CEGA.
>
> > **Reviewer**: The experiment used six common graph datasets. By comparing with benchmark methods such as Random, GRAIN, and AGE, the superiority of the CEGA method has been verified. The article evaluated the performance of various methods under different query constraints by testing different query budgets from 2C to 20C nodes. The article also conducted ablation experiments to analyze the contribution of each module to overall performance by removing certain evaluation modules of CEGA.
>
> **Authors**: We thank you for your detailed and accurate summary of our experimental setup. Your expertise-based recognition of our effort to maintain consistency with widely accepted common practice in the respective field is highly appreciated. As discussed with Reviewers oxao and JNMU, our evaluation step is carefully designed to align with standards adopted throughout GNN-based MEA literature. We will prepare further clear justification on the CEGA evaluation setup as suggested by Reviewers oxao and JNMU, and we are open to incorporating any additional suggestion you may have that could enhance the comprehensiveness of our evaluation.
>
> > **Reviewer**: Most existing research has overlooked budget constraints in practical applications, while CEGA proposes a more practical attack method by introducing limitations on the number of queries and query nodes per round. Current research mainly focuses on the overall information or local structure of the graph, while CEGA introduces a comprehensive consideration of historical information and structural centrality, making attacks more efficient on actual complex graph structures.
>
> **Authors**: We thank you for the effort you put into understanding CEGA's main contributions correctly. After a careful review of prior literature we cited in our paper, your comment captures our intent to fix a critical gap that remains in the GNN-based MEA literature by explicitly addressing realistic query budget constraints based on a history-based query-by-learning scheme while considering GNN-specific potential obstacles such as complexity on graph structure. As also noted in our discussion with Reviewer oxao, achieving consistently high performance across diverse datasets under such constraints remains a highly meaningful attempt for our community, and we are glad that your feedback naturally reveals your gentle but firm acknowledgment of such contribution.
>
> **Overall Responses by Authors**: We thank you once again for your thoughtful engagement and positive comments to our paper. We are fully committed to making the paper more solid and comprehensive in the next version as per the feedback received from Reviewers oxao and JNMU. We welcome any further discussion with you and will respond promptly to any of your follow-up questions.

---

### Decision · Program_Chairs · 2025-05-01

**Decision:**

Accept (poster)

**Comment:**

All reviewers positively evaluated the paper. For example, Reviewer JNMU states that the "proposed methodology is logically sound and well-suited for the problem" and "benchmark datasets represent diverse graph structures and applications, providing a robust testbed". The reviewers also acknowledge the importance of model extraction attacks, especially with a focus on practical scenarios with limited budget. For example, Reviewer E9DY praised the "new ideas for efficient and low-cost model extraction". Reviewer oxao initially has some concerns, but they increased their score after the rebuttal by the authors. Weaknesses raised by the other reviewers have either been sufficiently addressed or are only minor. Therefore, I recommend acceptance.